# Successful Utilization of Kidney Allografts with Diffuse Glomerular Fibrin Thrombi on the Preimplantation Biopsy after Circulatory Death: A Case Series

Sushma Bhusal [1], Hailey Hardgrave [2], Aparna Sharma [1], Lyle Burdine [3], Raj Patel [3], Gary Barone [3], Neriman Gokden [4] and Emmanouil Giorgakis [3,*]

1  Department of Medicine, Division of Nephrology, University of Arkansas for Medical Sciences, Little Rock, AR 72205, USA
2  Department of Surgery, University of Arkansas for Medical Sciences, Little Rock, AR 72205, USA
3  Department of Surgery, Transplant and Hepatobiliary Surgery, University of Arkansas for Medical Sciences, Little Rock, AR 72205, USA
4  Department of Pathology, University of Arkansas for Medical Sciences, Little Rock, AR 72205, USA
*  Correspondence: egiorgakis@uams.edu

**Abstract:** *Background:* Kidney allografts with the presence of diffuse glomerular fibrin thrombi are typically rejected by most centers due to concern for poor allograft outcomes in the recipients. The aim of this study was to report our single center experience in the use of such deceased donor allografts. *Methods:* Retrospective single-center cohort study of kidney transplant recipients who received deceased donor allografts with moderate-to-severe diffuse glomerular fibrin microthrombi on the pre-implantation biopsy. *Results:* Three adult recipients received deceased donor kidney transplantation from donation after circulatory death donors. One patient was pre-emptive to dialysis at the time of transplant. The donors had moderate-to-severe diffuse glomerular fibrin thrombi on preimplantation biopsies with no evidence of cortical necrosis. Mean follow-up period was 196 days. None of the recipients developed delayed allograft function. The mean 3-month and 6-month creatinine were 1.6 and 1.5 mg/dL, respectively, with corresponding mean eGFRs (estimated glomerular filtration rates) of 45.7 and 47.3 mL/min/1.73m$^2$. *Conclusions:* After excluding significant cortical necrosis by experienced transplant renal pathologist, otherwise transplantable kidney allografts with diffuse fibrin thrombi may be successfully transplanted in renal transplant recipients with good renal outcomes.

**Keywords:** fibrin thrombi; kidney; allograft

## 1. Introduction

Kidney transplant is the treatment of choice for end-stage renal disease (ESRD) patients. However, in the United States, about eighty thousand patients are on the kidney waitlist at any given time while half or less of those receive a transplant each year [1,2]. Given the scarcity of organs, it is imperative to minimize their inadvertent discard. Kidneys with diffuse glomerular fibrin thrombi (DGFT) are routinely declined by many transplant centers due to concerns over irreversible cortical damage due to glomerular thrombosis. Until recently, data on the outcomes of these kidneys were scant. The largest study was a retrospective, observational, single center study of 61 patients, which concluded that DGFT completely resolves soon after transplant, with no evidence that DGFT impacts long-term graft survival [3].

We present a case series of three kidney transplant recipients in whom donor kidneys with DGFT were successfully transplanted.

## 2. Patients and Methods

The study received institutional approval as a retrospective single-center case series of patients receiving deceased donor kidney transplant with preimplantation biopsies showing DFGT at our institution during period between 2018–2021. Follow-up period was through July 2022. Inclusion criteria: all deceased donor kidney transplant recipients with pathology evidence of DGFT at the preimplantation biopsy. Data were collected from single-surgeon prospectively populated electronic DGFT database (Epic Systems, 1979 Milky Way, Verona, WI, USA).

DGFT was defined by the presence of 50% or above of glomeruli-containing fibrin thrombi at the preimplantation biopsy. Delayed graft function was defined as the need for dialysis within the first week post-transplant. None of the allografts had been treated with fibrinolytic agents prior to implantation. No patient received systemic anticoagulation or fibrinolytic agents post-transplant. However, all recipients received low dose daily aspirin and received post-operative thromboprophylaxis while they were inpatients. All biopsy samples were wedge biopsies collected by the donor surgeon at the donor hospital at the time of organ procurement. All biopsy slides were reviewed by the same senior renal pathologist at the recipient hospital. We proceeded to transplant once cortical tubular coagulative necrosis more than >10% had been ruled out. One DGFT kidney was placed on pump during transfer to our center and while awaiting transplant. Follow-up biopsies were not performed on any of the recipients. Our center does not perform routine protocol biopsies.

All patients consented for receiving an acute kidney injury (AKI) and DGFT allograft after discussion with the transplant surgeon. All the recipients received standard immunosuppression with rabbit anti-thymocyte globulin and high dose methylprednisolone for induction followed by maintenance immunosuppression with tacrolimus (through goal 8–10 ng/dL in the first three months and 6-8 ng/dL after the third month) and mycophenolate 1 gm twice a day.

## 3. Results

*N* = 3 (Table 1). The recipients' mean age was 55, two were white and one was black, two were females. Two of the recipients (both females, aged 43 and 60), received mate kidneys from a single donor. Both kidneys had DGFT (>50% of glomeruli with thrombi) and were donations from circulatory death donors (DCDs). One recipient was pre-emptive to transplant.

**Table 1.** Donor characteristics and recipient follow-up.

| | COD | CIT | WIT (min) | DCD | eGFR (mL/min/1.73 m$^2$) | | Creatinine (mL/min/) | | DGF |
| --- | --- | --- | --- | --- | --- | --- | --- | --- | --- |
| | | | | | 3 Months | 6 Months | 3 Months | 6 Months | |
| Recipient 1 | TBI | 24 h 15 min | 22 | Yes | 52 | 57 | 1.2 | 1.3 | No |
| Recipient 2 | TBI | 28 h 20 min | 22 | Yes | 46 | 46 | 1.3 | 1.3 | No |
| Recipient 3 | ABI | 29 h 48 min | 28 | Yes | 39 | 39 | 2.1 | 1.8 | No |
| Mean | | 27 h 28 min | 24 | | 45.7 | 47.3 | 1.53 | 1.46 | |

ABI, anoxic brain injury; COD, cause of death; CIT, cold ischemia time; DCD, donation after circulatory death; TBI, traumatic brain injury; WIT, warm ischemia time; eGFR, estimated glomerular filtration rate.

All organs were from two DCD donors. The first donor was 27 years old and died from a gunshot injury to the head with subsequent traumatic brain injury and had disseminated intravascular coagulopathy (DIC). The first donor's warm ischemia time (WIT) was 22 min (Table 1). The second donor was 19 years of age with anoxic brain injury from a drug overdose, which was the cause of death. The second donor's WIT was 31 min (Table 1). Mean CIT was 27 h 28 min (range: 24 h 15 min–29 h 48 min).

The mean follow-up period was 196 days. None of the patients developed DGF. The mean 3-month and 6-month creatinine values were 1.6 and 1.5 mg/dL, respectively, with

corresponding mean estimated glomerular filtration rates (eGFR) of 46.5 and 46.4 mL/min/1.73 m$^2$. No patient required post-transplant biopsy. One recipient had BK viremia, treated with reduction in immunosuppression. Another recipient had refractory anemia stemming from Parvo virus B19 viremia, which improved after intravenous immunoglobulin infusions. All the patients maintained their renal function. No patient had rejection during the mean follow-up period.

## 4. Discussion

The dearth of kidney donations leads to a significant gap in the donations for the patients on the wait list [1,2]. Kidney allografts with moderate-to-severe DGFT are frequently declined by many centers.

In recent years, there has been a growing body of literature, albeit still few, which supports the use of these kidneys. One of the largest reported case series by Batra et al. compared 61 recipients with GFT and 557 controls. The study revealed the DGF rates was 49% in the GFT group and 39% in the control group [*p* = 0.14]. The serum creatinine and eGFR at 12 months were comparable in both groups. Estimated 1-year graft survival was also similar in both groups. The 1-month protocol biopsies performed on 52 patients in the GFT group (85%) revealed that only 4% (two biopsies) showed residual GFT. They concluded that GFT resolves rapidly after transplantation and it was safe to transplant selected kidneys with GFT [3]. McCall et al. reviewed 230 consecutive kidney biopsies, 8 of which exhibited microvascular thrombosis. The recipients of grafts with DGFT were at increased risk of developing DGF; however, graft function and survival at 1 and 2 years were comparable in both groups. Five of the eight recipients had follow-up biopsies, which showed no residual thrombi [4]. Soares et al. reported two successful transplants from brain dead donors with DIC, with preimplantation renal biopsies showing 100% GFT. Though both patients experienced DGF, they had a stable allograft function at 14 months post-transplant [5]. Similar reports of successful kidney transplants from donors with DIC and GFT have been published [6,7], including one in pediatric kidney transplant [8].

Even though, by convention, transplant surgeons shy away from donors with significant GFT, these are typically otherwise pristine, albeit AKI allografts, often from young donors who die from brain injury (as in our case series), resulting in DIC, which, in turn, cause DGFT. Prior experience from centers pioneering in the use of AKI allografts, including GFT kidneys, has shown that if the donor profile and biopsy are otherwise acceptable for transplant and there is no significant cortical necrosis verified by an experienced renal pathologist, these kidneys typically recover, with the fibrin clots being washed out in the early post-transplant period and AKI recovering in a pattern no different than non-GFT AKI allografts [3].

In our small series, one out of three patients was pre-emptive, and even though they all received DCD allografts with DGFT, none of them experienced DGF or required post-transplant for-cause biopsy. The key to successful transplantation of such organs is the reliable interpretation of the pre-implantation biopsy. Given the limitations inherent to the quality of the preimplantation biopsy fixation; staining; and, most importantly, biopsy interpretation at the donor hospital, which more often than not is performed under suboptimal conditions by pathologists not routinely engaged in the procurement of renal pathology frozen sections, it is crucial to have these slides reviewed by an experienced renal pathologist to reliably preclude significant cortical necrosis or other irreversible cortical damage/changes prior to committing to transplant. Furthermore, as it is always the case when it comes to transplanting AKI kidneys, the recipients should be appropriately educated and consenting of the possibility of DGF, and, of course, ischemic times should be kept at a minimum. Finally, if possible, such allografts would be preferable to be implanted to pre-emptive patients, who may afford slow allograft recovery without necessarily falling into DGF.

Our study is disadvantaged due to its small sample size and retrospective nature. There were no post-transplant protocol biopsies performed; however, the stable renal function in all the patients allows us to speculate that the thrombi resolved shortly after implantation.

In conclusion, donor kidneys with DGFT may yield comparable renal outcomes and should be considered for renal transplantation after appropriate review of the preimplantation biopsy to rule out cortical necrosis or other significant irreversible changes and after appropriate recipient consenting.

**Author Contributions:** Conceptualization, S.B. and E.G.; Methodology, S.B. and E.G.; Software, S.B. and E.G.; Validation, S.B., E.G., A.S., H.H., L.B., G.B., N.G. and R.P.; formal analysis, E.G.; investigation, S.B. and E.G.; resources, S.B.; data curation, S.B.; writing—original draft preparation, S.B.; writing—review and editing, S.B. and E.G.; visualization, S.B., E.G., A.S., H.H., L.B., G.B., N.G. and R.P.; supervision, E.G. All authors have read and agreed to the published version of the manuscript.

**Funding:** This research received no external funding.

**Institutional Review Board Statement:** The Institutional Review Board Director or Designee reviewed your material and determined that this project is NOT human subject research as defined in relevant federal regulations and/or UAMS IRB policy, and therefore it does not fall under the jurisdiction of the IRB review process.

**Informed Consent Statement:** Informed consent was obtained from all subjects involved in the study.

**Data Availability Statement:** All the data are available from the corresponding author upon reasonable request.

**Acknowledgments:** We acknowledge and thank all the health care professionals involved in the care of the patients and the patients for having tremendous patience and believing in us.

**Conflicts of Interest:** The authors declare no conflict of interest.

## Abbreviations

| | |
|---|---|
| AKI | Acute kidney injury |
| eGFR | estimated glomerular filtration rate |
| ESRD | End stage renal disease |
| DCD | Donation after Circulatory Death |
| DGF | Delayed allograft function |
| DIC | Disseminated Intravascular coagulation |
| GFT | Glomerular fibrin thrombi |
| CIT | Cold Ischemia Time |
| WIT | Warm Ischemia Time |
| COD | Cause of Death |
| TBI | Traumatic Brain Injury |

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
