# Peer review of "Successful Utilization of Kidney Allografts with Diffuse Glomerular Fibrin Thrombi on the Preimplantation Biopsy after Circulatory Death: A Case Series"

_2673-3943, doi:10.3390/transplantology4010002_

Round 1

Reviewer 1 Report

This paper entitled "Successful Utilization of Donation after Circulatory Death Kidney Allografts with Diffuse Glomerular Fibrin Thrombi on the Preimplantation Biopsy: A Case Series" shows the successful use of three kidneys from deceased donors showing diffuse glomerular thrombi at pre implantation biopsy.

Although overall presentation can be improved (i.e., making more clear clinical characteristics of each donor-recipient pair), the Authors show that adequate renal function recover was obtained in all three patient.

I have few major concerns

1) the study is fairly small compared to already available data: as the Authors acknowledge, a large 61-patient study from 2016 already showed usability of such patients

2) The most important clinical feature in these DCD transplant seems to be perfusion pressures data from machine perfusion performed before implantation. As of today, these are the most infromative data on the usability of such organs, and machine perfusion data are missing in the manuscript

Author Response

Taking the opportunity, the authors would like to thank the Editors and reviewer panel for their consideration.  The following are the responses to each point of concern by the reviewer. 

1) the study is fairly small compared to already available data: as the Authors acknowledge, a large 61-patient study from 2016 already showed usability of such patients

Indeed, this was a small single center case series. Given the rarity of reports on the use of such organs in the literature,  the authors think that our series would encourage more centers to use such organs under appropriate screening, with reasonable outcomes.

2) The most important clinical feature in these DCD transplants seems to be perfusion pressure data from machine perfusion performed before implantation. As of today, these are the most informative data on the usability of such organs, and machine perfusion data are missing in the manuscript. 

Respectfully, this is the reviewer’s personal view and is not supported by the literature. As per the literature cited in our report and our own experience, the major determinants of the successful utilization of such organs, are:

a)Validation of histopathological characteristics by a renal pathologist with prior experience in the interpretation of procurement frozen reads. It is key to be able to exclude  

i. Significant amount of cortical necrosis (>10%)

ii.    Rule out other common high pathological risk indicators: ie moderate/ severe glomerular sclerosis; arteriolar disease; IFTA; advanced chronic renal disease-specific findings

 iii.    Acceptable gross anatomical appearances

iv.    Keep CIT reasonably short

 v.    Avoid DCDs with WIT>30 min

b) Prudent donor-recipient match

c) Appropriate consenting of recipients, informing them the high possibility of transitional post-Tx HD until the organ recovers.

Once again, the authors would like to thank the Editors for their consideration. We hope that the manuscript in its current form meets the Journal’s publication standards and look forward to the next steps of the manuscript review.

S Bhusal MD

Reviewer 2 Report

The manuscript by Bhusal et al., reports three cases of end stage kidney disease pre-emptively transplanted with kidneys manifesting diffuse glomerular fibrin thrombi (DGFT) on preimplantation biopsies.  Two recipients received one kidney from the same donor, while the third received a kidney from a different donor.  None of the three recipients required dialysis post transplantation, and renal function was adequate at 3 and 6 months.   

Transplanting kidneys with DGFT has been amply reported before and thus this finding is not novel.  Furthermore neither is their finding of satisfactory renal function at 3 and 6 months post transplantation novel as outcomes after transplanting DGFT kidneys have previously been reported to be comparable to those of other deceased donor kidneys. 

What is relatively novel although not emphasized in this report, is that all three recipients were transplanted pre-emptively and consequently at least because of residual native renal function, but possibly due to the transplanted kidneys promptly commencing to function, none of the three recipients required dialysis and therefore, if dialysis  can be blamed for delaying recovery from acute kidney injury, these transplanted kidneys were spared such potential adverse effects.  In previous reports, 10-15% of recipients of DGFT kidneys had been pre-emptive, but little is reported about the need for dialysis in this special group of transplant recipients, except that in recipients of DGFT kidneys in general, the need for dialysis is somewhat greater than recipients of other deceased donor kidneys. Thus this

singular novel observation in the present report is not sufficiently highlighted or referenced by the authors .

The major criticism of the presented data is sparse information presented about the recipients.  Since all three recipients were pre-emptive, it is critical to know  what their pre-transplant eGFR's were, especially recognizing that the eGFR required for labelling an ESKD patient for pre-emptive transplantation varie and country.  If the authors can show graphically a downward trends of the serum creatinine in the short term after transplantation in each of the three recipients, it would be indicative of prompt function of the transplanted kidney.  

Minor Comment:

Please be consistent in using the abbreviation DGFT.

Minor English grammar and syntax issues should be addressed. 

Author Response

Taking the opportunity, the authors would like to thank the Editors and reviewer panel for their consideration.

  • Our original manuscript mentioned 2 out of 3 recipients were preemptive. However, upon further review, it was discovered that 1 of those 2 patients had started dialysis a couple of months prior to transplantation. The patient was not on dialysis at the initial evaluation and on follow-up visits during the waitlist. Eventually, there was a single preemptive patient. This has been revised.
  • The abbreviation DGFT has been used consistently in the revised manuscript. 

Once again, the authors would like to thank the Editors for the consideration. We hope that the manuscript in its current form meets the Journal’s  publication standards and look forward to the next steps of the manuscript review.

S Bhusal MD